# Hormonal Regulation and Transcriptomic Insights into Flower Development in *Hydrangea paniculata* ‘Vanilla Strawberry’

**DOI:** 10.3390/plants13040486

**Published:** 2024-02-08

**Authors:** Chao Xue, Yuxing Wen, Song Sheng, Yu Gao, Yaoyi Zhang, Tingfeng Chen, Jiqing Peng, Shoujin Cao

**Affiliations:** 1College of Forestry, Central South University of Forestry & Technology, 498 South Shaoshan Road, Changsha 410004, China; 13874803193@163.com (C.X.); wenyuxing2023@163.com (Y.W.); songsheng@csuft.edu.cn (S.S.); gy4732666@gmail.com (Y.G.); 15191249657@163.com (Y.Z.); 19118999801@163.com (T.C.); 2Yuelushan Laboratory, Hunan Agricultural University, Qiushi Building, Furong District, Changsha 410128, China; 3The Belt and Road International Union Research Center for Tropical Arid Non-Wood Forest in Hunan Province, 498 South Shaoshan Road, Changsha 410004, China

**Keywords:** hormone, regulation of flowering, transcriptome, molecular mechanism

## Abstract

Understanding the molecular mechanisms that regulate flower growth, development, and opening is of paramount importance, yet these processes remain less explored at the genetic level. Flower development in *Hydrangea paniculata* ‘Vanilla Strawberry’ is finely tuned through hormonal signals, yet the genetic underpinnings are not well defined. This study addresses the gap by examining the influence of gibberellic acid (GA3), salicylic acid (SA), and ethylene (ETH) on the flowering traits and underlying molecular responses. Treatment with 100 mg/L SA significantly improved chlorophyll content and bolstered the accumulation of soluble sugars and proteins, advancing the flowering onset by 6 days and lengthening the flowering period by 11 days. Concurrently, this treatment enhanced inflorescence dimensions, increasing length, width, and petal area by 22.76%, 26.74%, and 27.45%, respectively. Contrastingly, 100 mg/L GA3 expanded inflorescence size but postponed flowering initiation and decreased inflorescence count. Higher concentrations of SA and GA3, as well as any concentration of ETH, resulted in delayed flowering and inferior inflorescence attributes. A physiological analysis over 50 days revealed that these regulators variably affected sugar and protein levels and modified antioxidant enzyme activities. An RNA-seq analysis during floral development highlighted significant transcriptomic reprogramming, with SA treatment downregulating Myb transcription factors, implicating them in the modulation of flowering timing and stress adaptation. These findings illuminate the complex interplay between hormonal treatments, gene expression, and flowering phenotypes in *Hydrangea paniculata*, offering valuable perspectives for ornamental horticulture optimization.

## 1. Introduction

*Hydrangea paniculata*, commonly referred to as Panicle Hydrangea, is a deciduous shrub or small tree from the Saxifragaceae family, which has garnered significant attention in research due to its diverse applications and ornamental value. This species is celebrated for its extended and visually striking flowering phase, with some varieties exhibiting a flowering duration of over 200 days [1]. *Hydrangea paniculata* exhibits a range of physiological responses to environmental stressors, such as petroleum hydrocarbon contamination, indicating a spectrum of tolerance levels across different varieties [2]. Its resilience to challenging environmental conditions like drought highlights its potential for use in ecological landscaping and urban green spaces.

Innovations in propagation techniques, especially in vitro seed germination, have been crucial in the cultivation and genetic study of *Hydrangea paniculata*. Optimal media formulations have been identified to enhance germination rates, facilitating efficient propagation [3]. The use of plant growth regulators, such as dikegulac sodium, has been shown to improve branching without negatively impacting flower count, thus enhancing the ornamental appeal of *Hydrangea paniculata* [4]. However, the impact of certain growth regulators like IBA on in vitro rooting in related species has been minimal, suggesting a specificity in hormonal responses among hydrangea varieties [5]. Genetic analyses have provided insights into the relationships among hydrangea cultivars, offering opportunities for targeted breeding to enhance specific traits [6].

The focus on gibberellic acid (GA), salicylic acid (SA), and ethylene (ETH) in relation to flower development and characteristics is underpinned by their distinct roles in plant physiology and their practical applications in agriculture and horticulture. Gibberellic acid is crucial for promoting growth and development processes, including stem elongation, germination, and flowering, and is known for its ability to induce flowering and influence flower size and number [7,8]. Salicylic acid, primarily associated with plant defense mechanisms, also regulates physiological processes like flowering, influencing flowering time and response to environmental stress [9,10]. Ethylene, a gaseous hormone, plays a role in fruit ripening, flower senescence, and abscission, and affects sex determination in flowers [11]. These hormones are significant in agriculture and horticulture for their practical applications. GA is used to manipulate flowering times and improve flower quality, ETH is used in controlling fruit ripening and flower senescence, and SA in enhancing plant resilience and health [12,13]. The extensive research on GA, SA, and ETH provides a deep understanding of their mechanisms and effects, supporting their practical application in various agricultural and horticultural practices [14,15]. Their unique and diverse effects on plant physiology are not replicated by other plant hormones. For example, ethylene’s role in promoting flower senescence is unique [16]. GA, SA, and ETH interact with other hormones like auxins, cytokinins, and abscisic acid, regulating flowering in complex ways [17,18]. While other hormones also play significant roles in plant growth and development, the focus on GA, SA, and ETH is due to their direct and pronounced effects on flowering processes and their practical applications in agriculture and horticulture [19,20]. This study’s measurement of the effects of these plant growth regulators (PGRs) on the inflorescence characteristics of *Hydrangea paniculata*, such as size and number, is integral to understanding their influence on ornamental value and potential applications in enhancing floral traits for horticultural practices.

The application of gibberellic acid (GA3), salicylic acid (SA), and Ethephon has been extensively researched for their significant influence on flowering characteristics in various plant species, demonstrating their critical roles in horticultural and agricultural practices. GA3 and SA, known for their roles in growth and defense mechanisms, respectively, have been shown to enhance flower number, quality, vase life, and growth parameters such as plant height and inflorescence size. For example, GA3 or SA at 200 mg/L significantly increased flower numbers in dahlia and improved plant height and flowering time in African marigold and carnation varieties [21,22,23]. GA3 treatments also promoted early flower initiation and increased flower diameter and yield in carnation, as well as counteracting the inhibitory effects of abscisic acid and ethylene on flowering [24,25,26]. In contrast, Ethephon, a plant growth regulator known for its role in ethylene production, has been found to alter flower sex expression and fruit development. It increases the pistillate to staminate flower ratio in cucumber and squash, enhances flowering in apple seedlings, and can delay flowering and reduce plant height in various species [27,28,29,30]. These findings highlight the potential of GA3, SA, and Ethephon as tools in modulating flowering characteristics, offering valuable insights for enhancing flower production and quality, although their specific effects and optimal concentrations may vary depending on the plant species and environmental conditions.

In summary, the intricate dance of hormonal regulation in *Hydrangea paniculata* underscores a broader theme in plant biology: the delicate interplay between genetic predisposition and environmental modulation. The nuanced roles of gibberellic acid, salicylic acid, and ethylene in orchestrating the symphony of flowering processes highlight the complexity of plant developmental systems. These hormones, pivotal in their respective domains of influence, collectively contribute to a multifaceted regulatory network that governs flowering dynamics. This network, while responsive to external hormonal applications, is also deeply rooted in the plant’s inherent genetic framework. As we delve deeper into the genetic and hormonal underpinnings of flower development in *Hydrangea paniculata*, we stand on the cusp of unlocking new dimensions in horticultural science. This exploration not only promises enhanced aesthetic and commercial value for ornamental plants but also offers a window into the sophisticated mechanisms of plant growth and adaptation. The ensuing sections of this study aim to unravel these complexities, providing insights that could revolutionize our approach to plant cultivation and breeding in the ever-evolving landscape of agricultural science.

## 2. Results

### 2.1. Effect of Plant Growth Regulators on the Flowering Stage of Conical Hydrangea

The impact of plant growth regulators gibberellic acid (GA3), salicylic acid (SA), and ethylene (ETH) on the flowering period of *Hydrangea paniculata* was significant, as detailed in Table 1. GA3 treatments, across all concentrations, delayed the onset of flowering and shortened the total flowering duration, with higher concentrations exacerbating these effects. Notably, the 100 mg/L GA3 treatment did not significantly differ from the control in terms of the initial flowering period and total flowering duration, whereas the 500 and 1000 mg/L GA3 treatments markedly delayed the initial flowering and reduced the overall flowering period. For SA treatments, an increase in concentration led to a more pronounced delay in both the initial and peak flowering dates, along with a reduction in the total flowering duration. The 100 mg/L SA treatment, however, advanced the initial flowering by 5 ± 0.47 days and extended the total flowering duration by 10 days compared to the control, while higher concentrations (500 and 1000 mg/L) resulted in significant delays and reductions in flowering duration. ETH treatments, irrespective of concentration, significantly delayed the initial flowering period and shortened the total flowering duration, although the differences among the ETH treatments were not marked. In summary, while the 100 mg/L SA treatment effectively advanced the flowering period and extended its duration in *Hydrangea paniculata*, medium and high concentrations of SA, along with all concentrations of GA3 and ETH, delayed the flowering period and shortened its overall duration.

### 2.2. Effects of Plant Growth Regulators on the Inflorescence of Hydrangea paniculata

The effects of different concentrations of gibberellic acid (GA3), salicylic acid (SA), and ethylene (ETH) on the inflorescence of *Hydrangea paniculata* are shown in Table 2 and Appendix A. Under GA3 treatment, the length, width, and petal area of the inflorescence decreased with increasing concentrations, while the number of inflorescences increased. Specifically, the G1 treatment group showed significant increases in inflorescence length, width, and petal area by 7.54%, 12.04%, and 2.61%, respectively, compared to the control, but with a 25% reduction in the number of inflorescences. The G2 group showed no significant differences in these parameters compared to the control, while the G3 group exhibited significant reductions in inflorescence length, width, and petal area by 11.05%, 6.36%, and 33.33%, respectively, with no significant change in the number of inflorescences. In the SA treatment groups, the length, width, and petal area of the inflorescence decreased with increasing concentrations, but the number of inflorescences increased. The S1 group notably exceeded the control in inflorescence length, width, and petal area, showing increases of 22.76%, 26.74%, and 27.45%, respectively. The S2 and S3 groups had significantly lower values in these parameters compared to the control. For the ETH treatment groups, the length, width, and petal area of the inflorescence increased with higher concentrations, but the number of inflorescences noticeably decreased. The E1 and E2 groups had significantly lower values in these parameters compared to the control, but a significant increase in the number of inflorescences. The E3 group showed no significant differences in any of these parameters compared to the control. In summary, the 100 mg/L SA treatment was most effective in enhancing the quality of *Hydrangea paniculata* inflorescences, significantly increasing the length, width, and petal area. The 100 mg/L GA3 treatment significantly increased the length and width of the inflorescence without affecting the petal area, although it significantly reduced the number of inflorescences. The 500 mg/L GA3 and 150 mg/L ETH treatments showed no significant changes in inflorescence-related parameters. Other plant growth regulator treatments generally led to poorer inflorescence quality in *Hydrangea paniculata*.

### 2.3. Physiological Responses to Plant Growth Regulators in Hydrangea paniculata

Our research, encapsulated in Figure 1, provides a comprehensive analysis of how gibberellic acid (GA3), salicylic acid (SA), and ethylene (ETH) influence various physiological parameters in *Hydrangea paniculata* over a 50-day period. The study revealed that GA3 and ETH treatments predominantly increased soluble sugar levels, particularly in the later stages of the experiment, while SA treatments temporarily boosted soluble sugar content. In contrast, both SA and ETH treatments generally led to a reduction in soluble protein levels for most of the experimental duration. A notable finding was the tendency of all growth regulators to enhance malondialdehyde (MDA) accumulation, suggesting an increase in oxidative stress. However, this effect was somewhat mitigated in the later stages of GA3 and SA treatments. Additionally, antioxidant enzyme activities, specifically superoxide dismutase (SOD) and peroxidase (POD), were frequently elevated in response to the treatments, indicating an induced activation of antioxidant defenses. This increase, however, varied across different treatment groups and time points. The study also observed dynamic trends in the soluble sugar and protein content, with fluctuating patterns under natural conditions and following GA3 treatment, and varied responses under SA and ETH treatments. Overall, our findings demonstrate that the application of GA3, SA, and ETH significantly impacts the nutrient content, antioxidant enzyme activity, and MDA levels in *Hydrangea paniculata* leaves. These effects highlight the distinct and time-dependent influences of these plant growth regulators on the plant’s nutrient status, oxidative stress markers, and antioxidant capacity, providing valuable insights into their roles in plant physiology.

### 2.4. Library Construction and Transcriptome Sequencing

To further unravel the effect of SA at the molecular level, we utilized RNA sequencing (RNA-seq) to delve into the molecular mechanisms underlying the significant effects of SA on nutrient content, antioxidant enzyme activity, and malondialdehyde (MDA) levels in *Hydrangea paniculata* ‘Vanilla Strawberry’ leaves. The RNA-seq experiment was strategically designed to capture transcriptional changes during critical phases of flower bud differentiation. Specifically, on 23 May 2022 and 2 June 2022, corresponding to the near and the early stages of flower bud differentiation, we collected samples from both the SA treatment group and the control group. These were organized into four groups: SA1, SA2, CK1, and CK2, with SA1 and SA2 representing the near and the early stages of the 100 mg/L SA treatment, and CK1 and CK2 representing the control group at the same stages, each with three replicates.

To ensure the reliability of our RNA-seq data, we conducted a validation using quantitative PCR (qPCR) on nine randomly selected genes associated with flowering. The results demonstrated a high correlation with the RNA-seq data, affirming its accuracy, as depicted in Figure 2. The primers used are detailed in Appendix A. Quality control measures of the RNA-seq samples revealed excellent standards, including 41–55 million clean reads per sample, low error rates (0.0276–0.0286%), high Q20 and Q30 scores (≥96.68% and ≥90.91%, respectively), and normal GC content (45–46%), as shown in Appendix A. This comprehensive approach not only validated the RNA-seq data but also provided a robust foundation for subsequent analyses. These analyses aimed to unravel the transcriptional changes associated with the application of SA in *Hydrangea paniculata*, offering valuable insights into the molecular basis of the physiological changes observed in the leaves following treatment with plant growth regulators.

### 2.5. Transcriptomic Profiling Reveals Differential Gene Expression Dynamics in Flower Development and Opening

In the RNA-seq analysis, a gene expression matrix was constructed to compare different conditions—CK1, CK2, SA1, and SA2—across two developmental stages of 10 and 20 days. Figure 3A displays the count of upregulated and downregulated genes, along with the total DEGs, highlighting the dynamic changes in gene expression during flower development. This matrix reveals significant transcriptomic alterations, with CK1 and SA2 at 20 days showing a total of 4154 DEGs. Venn diagrams (Figure 3B–D) illustrate the overlap and uniqueness in gene expression profiles across different comparisons. A shared core of 480 genes was identified, suggesting the involvement of conserved genetic pathways in flower development. The analysis also delineates upregulated and downregulated DEGs, indicating condition-specific gene regulation. The enrichment analysis (Figure 3E) identifies genes involved in key biological processes such as carbohydrate metabolism and photosynthesis, highlighting the interaction between genetic regulation and environmental factors.

This finding reflects the complex interplay between genetic regulation and environmental factors, thereby emphasizing the multifaceted nature of flower development. In summary, this RNA-seq DEG analysis provides a panoramic view of the transcriptional landscape, offering invaluable insights into the genes and pathways fundamental to flower development and opening. These data not only propel our understanding of the genetic underpinnings of floral development forward but also lay the groundwork for future functional studies aimed at elucidating the precise roles of these DEGs in plant biology.

### 2.6. Elucidating Salicylic Acid-Induced Gene Expression Patterns in Flower Development: A Hierarchical Clustering Approach

The comprehensive RNA-seq analysis revealed a nuanced transcriptomic response to flower developmental cues and salicylic acid treatment. In the control condition (CK1), we observed an upregulation of a specific gene set, which notably showed a reciprocal downregulation in the SA2 condition, treated with 100 mg/L salicylic acid. This bidirectional shift in gene expression is characterized by a significant enrichment of Gene Ontology (GO) terms that are crucial for plant development and photosynthetic processes, including “carbohydrate metabolic process”, “photosynthesis”, “response to light stimulus”, “light harvesting in photosystem I”, and “chlorophyll biosynthetic process”. The enrichment in these GO terms provides a targeted framework for further investigation into the molecular mechanisms that are likely underpinning essential biological functions in flower development and plant growth.

To identify core candidate genes within this enriched transcriptomic landscape, we employed a hierarchical clustering algorithm on the RNA-seq data. By calculating Euclidean distances for measuring similarity between gene expression profiles and using Ward’s method for clustering, we achieved a classification based on minimal within-cluster variance. This approach effectively pinpointed clusters of genes with coordinated expression patterns, which were visualized through a dendrogram, elucidating their hierarchical relationships. This analytical structure was crucial in identifying a select group of candidate genes with significant expression changes, potentially playing key roles in critical physiological processes. Figure 4 presents the outcomes of our clustering analysis, delineating 50 distinct groups, each characterized by a unique expression signature. Notably, groups 15, 18, 32, 37, and 40, encompassing a collective total of 542 genes, approximately 11.56% of the 4686 genes analyzed, were selected for in-depth analysis. These groups were chosen based on their distinctive downregulation pattern in the SA2 condition, highlighting the specificity of their induction by SA and their potential significance in flower development and plant growth. This selection, rooted in the clustering analysis, was instrumental in isolating these genes for subsequent validation and functional characterization, paving the way for an in-depth exploration of their roles and mechanisms in response to SA treatment. This targeted analysis underscores the efficacy of an integrative transcriptomic approach, merging gene expression data with biological functionality to decode the complexities of plant developmental regulation. The insights gleaned from these candidate genes promise to enhance our understanding of the molecular interplay during flower development and underscore the role of SA in modulating plant physiological processes.

### 2.7. Regulatory Dynamics of Myb Transcription Factors in Plant Development and Stress Response

Our integrative Gene Ontology (GO) enrichment and Pfam domain analyses have illuminated the transcriptional complexity underpinning plant growth, development, and response to salicylic acid treatment. The GO analysis of upregulated DEGs and the 542 gene clusters, as illustrated in Figure 5, highlights a substantial enrichment of genes involved in photosynthesis, chlorophyll binding and biosynthesis, energy transfer, and metabolic processes—foundational components for plant vitality and development. This enrichment is further characterized by an array of GO terms related to the plant’s adaptive mechanisms to environmental stimuli such as light, reinforcing the integral role of these genes in facilitating the synthesis of essential biomolecules for plant growth.

Advancing into the molecular architects of these processes, our focus turned to transcription factors, specifically those with the Myb-like DNA-binding domain (Pfam: PF00249), which emerged prominently from our domain frequency analysis of the selected genes (Appendix A). Myb TFs, recognized for their regulatory versatility, were identified in nine genes, with eight representing novel Myb-like candidates, and one known Myb TF, DN1106_c0_g2 (Figure 6, Appendix A). This discovery suggests a broader spectrum of Myb TFs potentially involved in floral development than previously recognized. An expression analysis across conditions CK1, CK2, SA1, and SA2 revealed a significant downregulation of these Myb domain-containing genes in response to SA2, indicating a potential role in the negative regulation of growth pathways during high salicylic acid exposure. The most striking repression was observed in TRINITY_DN24371_c0_g2, which exhibited an almost complete suppression of expression in SA2, suggesting a robust regulatory mechanism at play. The magnitude of downregulation, represented by the depth of green shading in the expression data, underscores the sensitivity of these TFs to salicylic acid, positioning them as key elements in the plant’s stress response repertoire.

The combined data present a comprehensive portrait of gene regulation where Myb TFs are likely critical players in modulating the plant’s developmental and stress response pathways. The notable downregulation of these genes, particularly under stress conditions, opens new avenues for understanding the intricate relationship between transcriptional regulation and plant adaptive responses. TRINITY_DN24371_c0_g2 stands out among these for its dramatic response to salicylic acid treatment, warranting further exploration to decipher its exact role in plant development and stress response. Our findings not only enrich the current understanding of the transcriptional network in plants but also set a precedent for the functional characterization of Myb TFs in plant biology.

## 3. Discussion

The foliar application of hormones in plant development is a critical aspect of modern horticulture and agriculture, offering a nuanced approach to influencing plant growth and flowering. This technique, which involves the direct application of hormones to the foliage, has a broad-spectrum impact on plant growth and development. Hormones such as brassinosteroids, gibberellic acid (GA3), salicylic acid (SA), and ethylene (ETH) are known for their profound effects on various physiological processes. For instance, brassinosteroids have been observed to significantly enhance leaf area index, seedling height, biomass, and enzyme activities in pistachio seedlings [31]. This demonstrates their crucial role not only in promoting growth but also in enhancing the plant’s physiological capabilities. Similarly, the application of tripotassium phosphate and plant exogenous hormone treatments can lead to alterations in gene expression levels, thereby influencing plant growth and development in species like pomegranate [32]. These findings underscore the potential of foliar hormone application in manipulating the genetic and metabolic pathways of plants for improved growth outcomes.

The influence of hormone application extends specifically to flower development, a critical aspect for both ornamental and fruit-bearing plants. The application of cytokinins, for example, has been shown to significantly increase the total number of flowers in jojoba plants [33], highlighting its role in enhancing floral proliferation. Additionally, hormones like GA3, IBA, and NAA have been found to significantly affect the growth, flowering, and quality of spikes in daisy, indicating the diverse impacts of different hormones on flowering processes. Notably, the application of GA3 at specific concentrations during the pre-blooming stage can improve flowering, fruit set, and fruit retention in crops like cashew [34], demonstrating its utility in enhancing reproductive success and yield. These findings suggest that the strategic application of hormones can be a powerful tool in optimizing flowering and fruiting in various plant species. Moreover, hormone application plays a crucial role in enhancing flower growth and fruit quality. Studies have shown that hormone treatments not only promote flower growth but also improve the quality of fruit crops [35]. In *Hydrangea paniculata*, for instance, the application of GA3, SA, and ETH significantly influences nutrient content, antioxidant enzyme activity, and MDA levels. This indicates that hormone treatments can enhance the plant’s nutritional status and stress resilience, thereby positively impacting flower growth and development. The ability of hormones to modulate nutrient content and stress responses in plants opens up new avenues for improving plant health and productivity through foliar applications. In conclusion, the foliar application of hormones represents a sophisticated and effective approach to plant management, offering significant benefits in terms of growth enhancement, flowering optimization, and fruit quality improvement. The ability of these treatments to influence physiological and molecular processes in plants underscores their importance in modern agricultural and horticultural practices. As research continues to unravel the complex interactions between hormone treatments and plant development, the potential for refined and targeted applications of these substances in plant cultivation is likely to expand, offering exciting prospects for the future of plant science and agriculture.

The distinct effects of plant growth regulators (PGRs) such as gibberellic acid (GA3) and salicylic acid (SA) on flowering and inflorescence are pivotal in understanding plant developmental biology. These regulators, when applied, have been observed to induce significant changes in plant morphology and physiology, leading to varied responses in different plant species. For instance, GA3 and SA applications have resulted in increased plant height, reduced days to flowering, and elongated inflorescence in species like Limonium var. Misty Blue [36]. This is indicative of the role that these hormones play in accelerating growth and developmental processes. In the case of Zantedeschia, GA3 has been shown to prolong the flowering period and increase the number of flowering shoots [37], a finding that is mirrored in *Hydrangea paniculata* where GA3 treatments delayed the onset of flowering. These observations suggest that GA3 not only promotes growth but also modulates the timing of reproductive phases.

The physiological and molecular responses to these growth regulators further elucidate their roles in plant development. The application of GA3, SA, and ETH leads to alterations in the plant’s metabolic profile, as evidenced by changes in soluble sugars, starch, soluble protein, malondialdehyde (MDA) content, and antioxidant enzyme activities. These changes are reflective of the plant’s adaptive mechanisms to the hormonal treatments, adjusting its metabolic pathways and stress responses accordingly. For example, the application of GA3 in Phalaenopsis amabilis led to an increase in sugar content in both leaves and inflorescence, effectively reversing blocked flowering [38,39]. Similarly, GA3 treatment in strawberry plants resulted in accelerated flowering and an increase in the number of flower buds and open flowers [40]. These instances highlight the profound impact of PGRs on the plant’s physiological state, influencing not only growth patterns but also the timing and intensity of flowering.

The intricate interplay between hormonal treatments and their consequent physiological and molecular changes in *Hydrangea paniculata* ‘Vanilla Strawberry’ is a testament to the complexity of plant developmental processes. This study illuminates how plant growth regulators (PGRs) such as gibberellic acid (GA3), salicylic acid (SA), and ethylene (ETH) orchestrate a symphony of signaling pathways and gene expression modifications, culminating in discernible alterations in plant morphology and development. The diverse responses elicited by these PGRs underscore their potential as pivotal tools in horticulture and agriculture, facilitating crop improvement and management. Delving into the mechanisms behind these responses not only enriches our understanding of plant biology but also paves the way for the precise manipulation of plant growth and development, thereby enhancing agricultural productivity and efficiency [41,42]. This research provides a comprehensive view of how GA3, SA, and ETH influence the phenotypic traits of flowering in *Hydrangea paniculata*, as well as the underlying molecular responses. Notably, the application of 100 mg/L SA demonstrated significant effects on the flowering process, enhancing chlorophyll content and boosting the accumulation of soluble sugars and proteins. These biochemical enhancements led to an earlier onset of flowering and an extended flowering period, traits highly desirable in ornamental horticulture. The increase in inflorescence dimensions under SA treatment underscores its efficacy in modifying floral characteristics for aesthetic and commercial purposes [43]. Conversely, GA3 application exhibited a distinct set of effects. While it increased inflorescence size, it also delayed flowering initiation and reduced the number of inflorescences, suggesting a nuanced role in floral regulation. The differential responses to varying concentrations of SA and GA3, along with the effects of ETH, highlight the need for a balanced and specific approach in hormonal treatments to achieve desired outcomes in flower development [44,45]. The physiological analysis over a 50-day period revealed dynamic changes in sugar and protein levels and modifications in antioxidant enzyme activities, indicative of the plant’s adaptive responses to hormonal treatments. These changes reflect alterations in metabolic processes and stress resilience mechanisms. The RNA-seq analysis during floral development further emphasized significant transcriptomic reprogramming in response to these hormonal treatments. The downregulation of Myb transcription factors in response to SA treatment is particularly noteworthy, as it implicates these factors in the modulation of flowering timing and stress adaptation, opening new avenues for understanding the genetic regulation of flowering and stress responses in plants [46,47].

Our study on the distinctive reactions of *Hydrangea paniculata* ‘Vanilla Strawberry’ to gibberellic acid (GA3), salicylic acid (SA), and ethylene (ETH) suggests that these responses may be genetically and physiologically distinct from those of other species. Based on their distinct genetic composition and interactions with their surroundings, plants may respond differently to hormone therapies depending on the species [48]. For instance, while GA3 typically promotes flowering in many species [49], its effect on delaying flowering in our study suggests a distinct genetic response in *Hydrangea paniculata*, possibly due to its unique gene expression patterns or hormonal interaction networks [50]. Moreover, regarding the regulation of flowering in plants, it is well established that a complex network of genes orchestrates this process. The MYB gene family, known for its role in various plant processes, including flowering, is a key player in this network [51]. Our findings indicate that salicylic acid treatment led to the downregulation of MYB transcription factors, suggesting their involvement in flowering timing and stress adaptation in *Hydrangea paniculata*. This aligns with studies in other species where MYB genes have been implicated in similar roles, indicating a conserved function across different plant taxa [52]. However, the specific roles and mechanisms of MYB genes can vary, reflecting the diversity in flowering regulation strategies among different plant species [53].

In summary, this study sheds light on the complex interplay between hormonal treatments, gene expression, and flowering phenotypes in *Hydrangea paniculata*. The insights gained not only enhance our understanding of plant developmental biology but also offer valuable perspectives for optimizing ornamental horticulture. By manipulating hormonal treatments, it is possible to tailor flowering traits to specific aesthetic and commercial needs, advancing the field of plant cultivation and breeding. Future research, particularly focusing on the functional characterization of key genes and the interaction of hormonal signals with environmental factors, will further refine our ability to control and enhance flowering in ornamental plants.

## 4. Materials and Methods

### 4.1. Plant Material

The study utilized four-year-old *Hydrangea paniculata* ‘Vanilla Strawberry’ plants, characterized by robust growth and uniformity in height and crown width, averaging 56.0 cm and 52.6 cm, respectively. Each plant was potted in a plastic container (39.5 cm diameter × 22.5 cm base diameter × 29.8 cm height). The cultivation substrate comprised a 1:1 mixture of local red soil (pH 5.8~6.0) and peat soil. Plants were placed outdoors with an 80 cm × 80 cm spacing and received standard care throughout the growth period.

### 4.2. Plant Material Treatment and Sample Collection

Following protocols from the relevant literature [54], the experiment employed three plant growth regulators: GA3 (100, 500, 1000 mg/L), SA (100, 500, 1000 mg/L), and ETH (25, 75, 150 mg/L), sourced from Beijing Solarbio Science & Technology Co., Ltd. (Beijing, China). Each regulator was applied in three concentrations, with sterile water as a control, forming 10 treatment groups (Appendix A), each with 10 pots and replicated three times. Starting on 29 April 2022, plants in the vegetative growth stage were sprayed every 7 days, totaling three applications. Leaf samples were collected starting on the 10th day post final spraying (23 May 2022), every 10 days for five collections (Figure 7). Flower bud development was documented using camera and paraffin section techniques. Leaves (3rd to 5th from the top) without pests or diseases were selected for physiological measurements. Flower bud samples from the 10th and 20th days were used for transcriptomic sequencing. Samples were rinsed, dried, frozen in liquid nitrogen, and stored at −80 °C, with three replicates each.

### 4.3. Flowering Index Measurement

In order to study the effects of hormone types and concentrations on the flowering indices of *Hydrangea paniculata*, the flowering indices were measured following previous research [55,56]. Observations included the beginning, peak, and end of the flowering period, with calculations for total flowering days, days advanced for initial flowering, and extended flowering duration. Total flowering days are the number of days each plant experienced from initial flowering to the end of flowering, that is, the end flowering date minus the initial flowering date. Days advanced for initial flowering are calculated as the initial flowering date of the treatment group minus the average initial flowering date of the control group. The extended flowering duration is the flowering period of the treatment group minus the average flowering period of the control group. The number of inflorescences in the treatment group and the control group were recorded, the length and width of inflorescences in full-bloom stage were measured with vernier calipers, the area of sepals was calculated with square paper, and the differences between the treatment group and the control group were analyzed with Excel version 2016.

### 4.4. Physiological Index Measurement

To investigate the effects of exogenous hormones on nutrients and resistance in *Hydrangea paniculata* ‘Vanilla Strawberry’, measurements of soluble sugars (SS), soluble starch (SST), soluble proteins (SP), malondialdehyde (MDA), superoxide dismutase (SOD), and peroxidase (POD) were conducted as per Ling et al. [57].

### 4.5. RNA Extraction, Testing, and Transcriptome Sequencing

The total RNA of flower bud samples from the SA treatment (early and late stages) and control group (early and late stages) was extracted using the RNA prep Pure polysaccharide and polyphenol plant total RNA extraction kit (Tiangen Biochemical Technology Co., Ltd., Beijing, China), according to the manufacturer’s protocol. The purity and concentration of the total RNA were detected using 1% agarose gel electrophoresis. The concentration of the total RNA was confirmed using the NanoDrop 2000 ultra-micro spectrophotometer, and the purity at OD260/OD280. The extracted total RNA was reverse-transcribed using the HiScript II Q RT SuperMix for qPCR (+gDNA wiper) reverse transcription kit (Nanjing Novizan Biotechnology Co., Ltd., Nanjing, China), in accordance with the manufacturer’s instructions. mRNA was separated from the total RNA using magnetic beads with Oligo (dT), and fragmentation buffer was added. mRNA was divided into short fragments randomly. cDNA was synthesized by 6-base random hexamers based on the mRNA template. Then, the buffer, dNTPs, DNA polymerase I, and RNase H were added to synthesize the double-stranded cDNA, and the double-stranded cDNA was purified using AMPure XP beads. The purified double-stranded cDNA was first end-repaired, the tail containing A was added, and the sequencing joint was connected, and then fragments of different sizes were screened using AMPure XP beads. Finally, PCR was amplified and the PCR products were purified using AMPure XP beads to obtain the final library. The cDNA was sequenced at Shanghai Meiji Biomedical Technology Co., Ltd. (Shanghai, China) using the Illumina platform.

### 4.6. Data Assembly and Bioinformatics Analysis

Raw reads were filtered to remove adapter sequences and low-quality reads, obtaining high-quality clean reads. Trinity assembly software (version 2.15.1) [58] was used for de novo assembly. To ensure assembly quality, homologous transcript clustering and sequence clustering software (RapClust (version 0.1.2)) were used for further splicing and redundancy removal, obtaining as long non-redundant unigenes as possible. Bioinformatics analysis included unigenes functional annotation (annotating unigenes to databases such as NCBI-NR (NCBI Non-redundant Protein Sequences Database), Swiss-prot (A Manually Annotated and Reviewed Protein Sequence Database), Pfam (Protein Family), eggNOG (COG: Clusters of Orthologous Groups of Proteins; KOG: euKaryotic Ortholog Groups), GO (Gene Ontology), and KEGG (Kyoto Encyclopedia of Genes and Genomes)), and differential gene expression analysis; DEGs were screened based on *p*-adjust < 0.05, |FC| ≥ 2, and subjected to enrichment analysis using KOBAS2.0 software [59].

### 4.7. qPCR Validation

To validate the accuracy of the transcriptome data, nine flowering-related genes were randomly selected based on sequencing results. Primers were designed online through NCBI and synthesized by Jingkairui Biotechnology Co., Wuhan, China. β-Action was used as the reference gene for qPCR experiments [60]. RNA returned by Shanghai Meiji Biomedical Co. was reverse-transcribed using the HiScript^®^ II Q RT SuperMix for qPCR (+gDNA wiper) kit to obtain cDNA, and the expression of selected genes was verified using the ChamQ Universal SYBR qPCR Master Mix kit, strictly following the manual instructions. Real-time fluorescence quantitative qPCR was performed on the CFX96 TouchTM Real-Time PCR Detection System (Bio-Rad, Hercules, CA, USA), and gene expression levels were calculated using the 2^−ΔΔCT^ formula [61].

### 4.8. Data Processing

Experimental data were organized using Microsoft Excel 2016. Univariate analysis and correlation analysis were performed using IBM SPSS Statistics 26.0 software, and graphs were drawn using OriginPro 2021.

## Figures and Tables

**Figure 1 plants-13-00486-f001:**
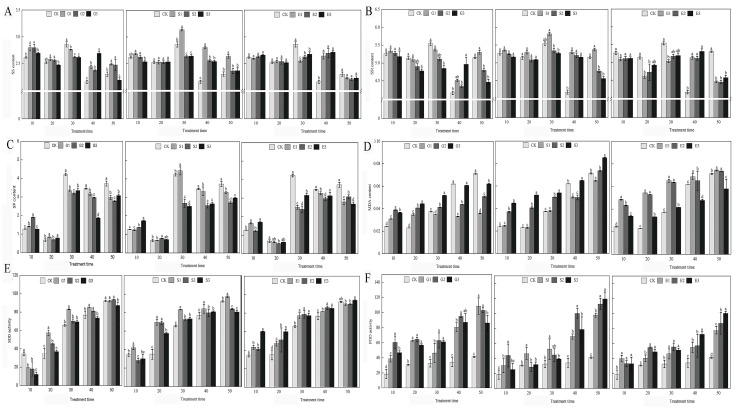
The effects of GA3, SA, and ETH on the nutrient content, antioxidant enzymes, and MDA of *Hydrangea paniculata* ‘Vanilla Strawberry’ leaves. (**A**–**C**) are the effects of GA3, SA, and ETH on soluble sugar (SS), soluble starch (SSt), and soluble protein (SP) of *Hydrangea paniculata* ‘Vanilla Strawberry’, respectively; (**D**–**F**) are the effects of GA3, SA, and ETH on MDA, SOD, and POD of *Hydrangea paniculata* ‘Vanilla Strawberry’, respectively. The different letters indicate significant difference among treatments (*p* ≤ 0.05).

**Figure 2 plants-13-00486-f002:**
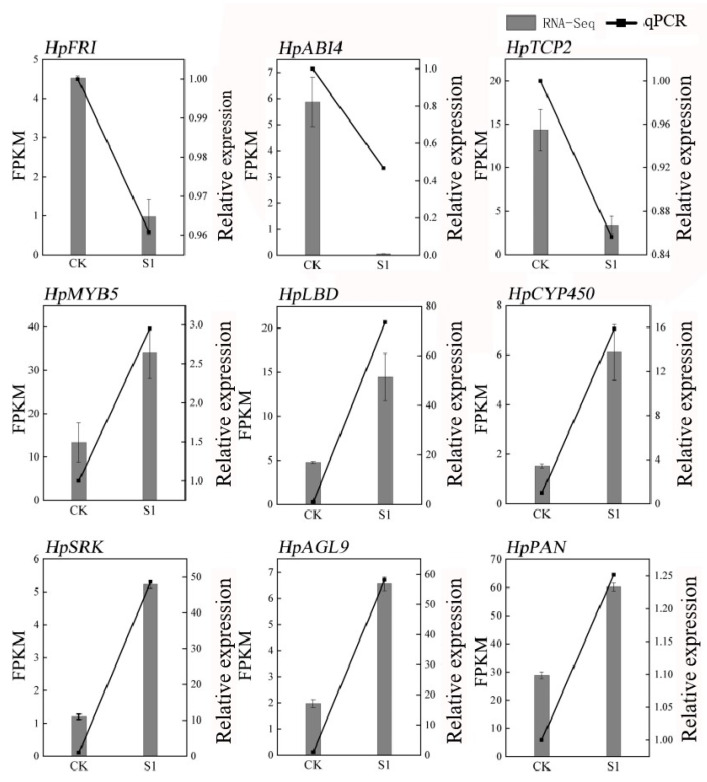
qPCR validation of RNA-seq data in *Hydrangea paniculata* ‘Vanilla Strawberry’. The nine genes in the figure are nine flowering-related genes randomly selected from the transcriptome database.

**Figure 3 plants-13-00486-f003:**
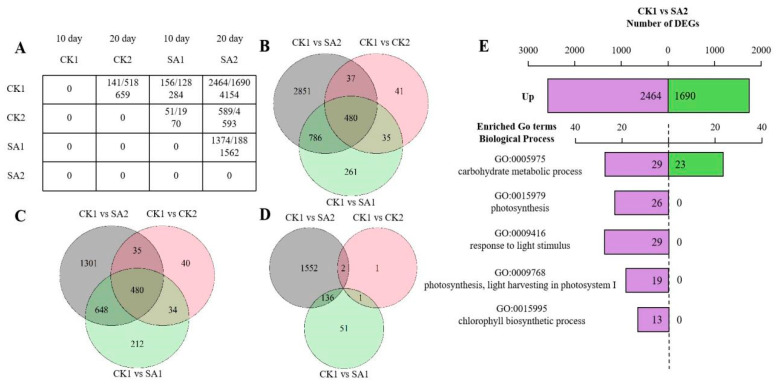
DEG analysis of RNA-seq of *Hydrangea paniculata* ‘Vanilla Strawberry’. (**A**) Matrix of differentially expressed genes (DEGs) across the treatment group (100 mg/L SA) and the control group (sterile water) at 10 days and 20 days. (**B**) Venn diagrams of all differentially expressed genes (DEGs) across different comparisons: CK1 versus SA2, CK1 versus CK2, and CK1 versus SA1. (**C**,**D**) Venn diagrams of upregulated and downregulated differentially expressed genes (DEGs) across different comparisons: CK1 versus SA2, CK1 versus CK2, and CK1 versus SA1. (**E**) KEGG enrichment analysis for the CK1 versus SA2 comparison. Purple represents the number of upregulated differentially expressed genes (DEGs) and green represents the number of downregulated DEGs.

**Figure 4 plants-13-00486-f004:**
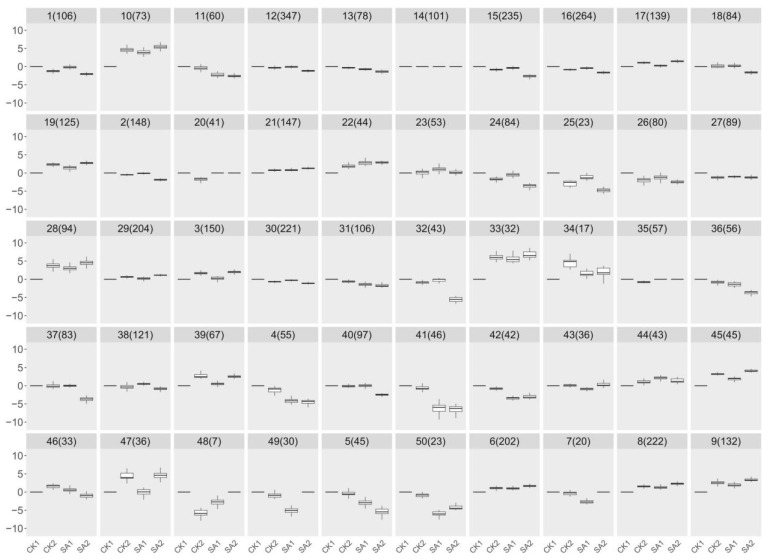
Analysis of RNA-seq using a hierarchical clustering algorithm. By calculating Euclidean distances for measuring similarity between gene expression profiles and using Ward’s method for clustering, 50 distinct groups were delineated based on minimal within-cluster variance.

**Figure 5 plants-13-00486-f005:**
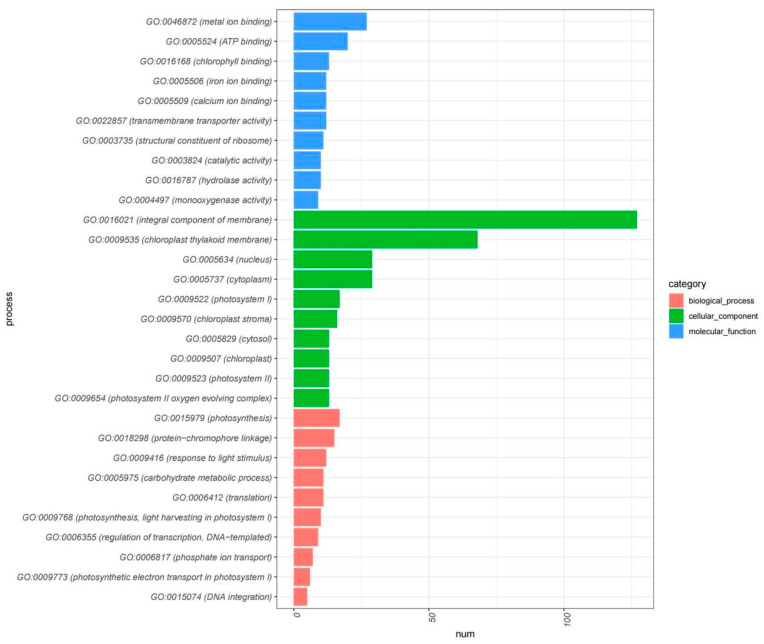
GO enrichment analysis of 542 core negative regulated genes.

**Figure 6 plants-13-00486-f006:**
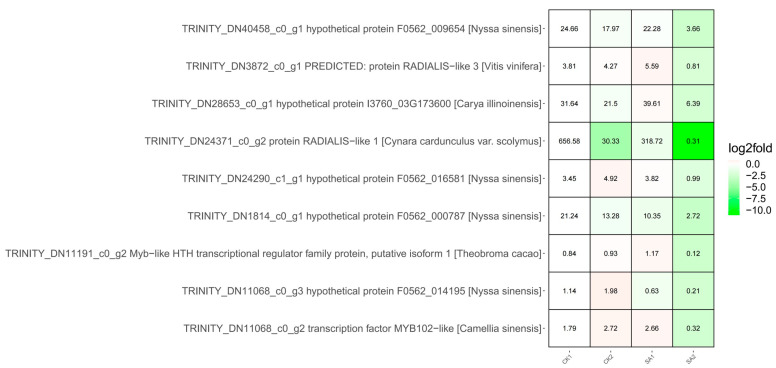
Co-transcription factors of the nine identified MYB genes. The color scale represents values of log2fold. The magnitude of downregulation is represented by the depth of green shading in the expression data. Dark green represents a larger downregulation, and light green represents a smaller downregulation.

**Figure 7 plants-13-00486-f007:**
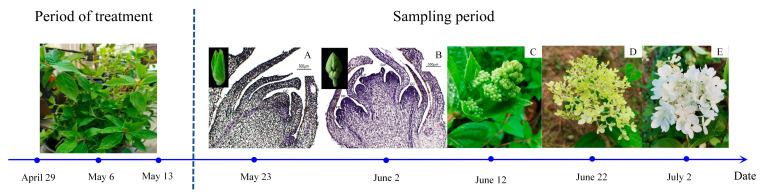
Division of flower bud (inflorescence) growth and development stages of *Hydrangea paniculata* ‘Vanilla Strawberry’ during the sampling period. (**A**–**E**) represent normal growth conditions, and the cone in the stages of flower bud differentiation: (**A**) near the differentiation stage of flower buds; (**B**) flower bud differentiation stage; (**C**) late stage of flower bud differentiation; (**D**) initial flowering stage; and (**E**) full-bloom stage.

**Table 1 plants-13-00486-t001:** Different concentrations of plant growth regulators and their effects on the flowering period of *Hydrangea paniculata* ‘Vanilla Strawberry’.

Treatment	Initial Flowering Date	Full Flowering Date	Final Flowering Date	Days Advanced for Initial Flowering (d)	Increased Days of Total Flowering Duration (d)
CK	11 June 2021	18 June 2021	29 August 2021	0 ± 1.89 ^b^	0 ± 1.70 ^b^
G1	14 June 2021	21 June 2021	29 August 2021	−3 ± 1.41 ^b^	−3 ± 1.24 ^b^
G2	20 June 2021	28 June 2021	27 August 2021	−9 ± 0.67 ^d^	−11 ± 0.82 ^c^
G3	23 June 2021	30 June 2021	19 August 2021	−12 ± 1.76 ^e^	−21 ± 1.49 ^d^
S1	6 June 2021	12 June 2021	4 September 2021	5 ± 0.47 ^a^	10 ± 0.82 ^a^
S2	17 June 2021	25 June 2021	23 August 2021	−6 ± 1.15 ^c^	−11 ± 0.99 ^c^
S3	20 June 2021	27 June 2021	20 August 2021	−9 ± 0.94 ^d^	−17 ± 1.05 ^e^
E1	21 June 2021	27 June 2021	26 August 2021	−10 ± 0.67 ^de^	−12 ± 0.82 ^c^
E2	22 June 2021	30 June 2021	27 August 2021	−11 ± 0.82 ^de^	−12 ± 1.05 ^c^
E3	20 June 2021	27 June 2021	26 August 2021	−9 ± 1.15 ^d^	−11 ± 1.25 ^c^

Note: Values are expressed as mean ± standard deviation. The different letters indicate significant difference among treatments (*p* ≤ 0.05).

**Table 2 plants-13-00486-t002:** Characterization of inflorescences at the full flowering stage of *Hydrangea paniculata* ‘Vanilla Strawberry’ under different treatment conditions.

Treatment	Inflorescence Length (mm)	Inflorescence Width (mm)	Number of Inflorescences	Petal Area (cm^2^)
CK	123.67 ± 1.25 ^c^	99.67 ± 2.05 ^c^	12 ± 0.99 ^d^	1.53 ± 0.02 ^b^
G1	133.00 ± 2.16 ^b^	111.67 ± 2.87 ^b^	9 ± 0.82 ^e^	1.57 ± 0.11 ^b^
G2	118.67 ± 4.50 ^cd^	97.33 ± 4.71 ^cd^	10 ± 1.15 ^de^	1.42 ± 0.09 ^b^
G3	110.00 ± 8.16 ^d^	93.33 ± 2.49 ^d^	12 ± 1.15 ^d^	1.02 ± 0.04 ^c^
S1	151.67 ± 2.36 ^a^	126.33 ± 1.25 ^a^	12 ± 0.82 ^d^	1.95 ± 0.08 ^a^
S2	113.00 ± 2.16 ^d^	91.67 ± 2.36 ^d^	13 ± 1.05 ^d^	1.12 ± 0.03 ^c^
S3	94.00 ± 0.94 ^e^	71.00 ± 1.41 ^f^	15 ± 0.47 ^c^	0.65 ± 0.03 ^e^
E1	92.00 ± 2.45 ^e^	81.67 ± 2.36 ^e^	19 ± 0.82 ^a^	0.75 ± 0.02 ^de^
E2	93.33 ± 2.49 ^e^	85.67 ± 4.19 ^e^	17 ± 1.05 ^b^	0.87 ± 0.03 ^d^
E3	119.37 ± 4.19 ^cd^	98.67 ± 1.89 ^cd^	12 ± 1.15 ^d^	1.43 ± 0.09 ^b^

Note: Values are expressed as mean ± standard deviation. The different letters indicate significant difference among treatments (*p* ≤ 0.05).

## Data Availability

The datasets presented in this study can be found in online repositories. The names of the repository/repositories and accession number(s) can be found below: https://www.ncbi.nlm.nih.gov/, PRJNA1057471.

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
