# Peer review of "Hormonal Regulation and Transcriptomic Insights into Flower Development in *Hydrangea paniculata* ‘Vanilla Strawberry’"

_plants, 2024, doi:10.3390/plants13040486_

Round 1
Reviewer 1 Report
Comments and Suggestions for Authors
The manuscript, “Hormonal Regulation and Transcriptomic Insights into Flower Development in Hydrangea paniculata 'Vanilla Strawberry'” details the morphological and gene-expression changes in response to differing applications of hormones. Lack of detail and clarity in Materials and Methods, all figure captions should be expanded to reflect the content and stand alone from the text. The Discussion needs major work to highlight the significant findings from this study and place them in the context of what is known in the field. Comments by line item are below.
The Introduction needs to contain clearly stated objectives.
136-140: The lettering on C – E is not visible
142: What are the DUS test guidelines for hydrangea? Please provide a citation.
144: Please provide more detail of the calculations here - early flowering advancement, and extended flowering duration.
147-150: What are the purpose of these measurements? Perhaps include in the introduction or discussion the reason for including these.
151 – 154: Even though this work was done by a company, you still need to report the methods and software used for RNA extraction, QC, and library construction.
161 – 164: Provide more details on the software used for these analyses.
200 – 202: Please include the number of plants observed for each treatment. Days and total flowering duration should be expressed as a mean +/- a standard deviation based on the number of plants observed.
231-233: Please include the number of inflorescences observed for each treatment. All variables should be expressed as a mean +/- a standard deviation based on the number of inflorescences or petals observed.
234-256: Comprehensive analysis, valuable insights- these are terms for the Discussion, not the results.
264: Why did you choose SA?
274: Which nine genes? This section on qPCR should follow RNA-seq analysis, as these nine genes have not yet been detailed.
290 – 323: This section is not appropriate for the Results; please move to the discussion. The Results should present the research findings without interpretation, supported by tables or figures.
324: Figure captions should stand alone. Always include the species, and any information needed to interpret the figure apart from the text. Panels A though E should be labeled in this caption.
410-516: This Discussion is a repeat of the introduction. Use this section to place your results in the context of the literature. For example, Why did you flowers exhibit opposite responses to what the literature reports? What about flowering regulation in other species? What genes are involved? Is the MYB gene you found similar to those?
Reviewer 2 Report
Comments and Suggestions for Authors
This very important study that which show the relationship between gene expression and flowering phenology. The results are of importance for commercial production and are beneficial for of plant cultivation and breeding of ornamental plants.
The paper is of high quality, all study questions and expectations are properly addressed. The methods are adequate . The discussion is based on obtained results .
I found this research and data analyses as very good and I found only very minor issue to correct before publication.
Table 1, Table 2 – add the cultivar name to the plant name – should be Hydrangea paniculate 'Vanilla Strawberry'
Figure 2 - add the cultivar name to the plant name – should be Hydrangea paniculate 'Vanilla Strawberry'
Round 2
Reviewer 1 Report
Comments and Suggestions for Authors
The authors addressed all mandatory comments which greatly improved the materials and methods, results, and presentation of the figures. Accept in present form.